# The Role of Gamma Oscillations in the Pathophysiology of Substance Use Disorders

**DOI:** 10.3390/jpm11010017

**Published:** 2020-12-28

**Authors:** Jessica U. Ramlakhan, Ming Ma, Reza Zomorrodi, Daniel M. Blumberger, Yoshihiro Noda, Mera S. Barr

**Affiliations:** 1Temerty Centre for Therapeutic Brain Intervention, Division of Mood and Anxiety, Centre for Addiction and Mental Health, Toronto, ON M6J1H4, Canada; jessica.ramlakhan@gmail.com (J.U.R.); mingdonna.ma@mail.utoronto.ca (M.M.); Reza.Zomorrodi@camh.ca (R.Z.); Daniel.Blumberger@camh.ca (D.M.B.); 2Institute of Medical Science, Faculty of Medicine, University of Toronto, Toronto, ON M5S1A8, Canada; 3Department of Psychiatry, Temerty Faculty of Medicine, University of Toronto, Toronto, ON M5S1A8, Canada; 4Multidisciplinary Translational Research Lab, Department of Neuropsychiatry, Keio University School of Medicine, Tokyo 160-8582, Japan; yoshi-tms@keio.jp

**Keywords:** gamma oscillations, alcohol, tobacco, cannabis, cognition, co-morbidity

## Abstract

Substance use disorders (SUDs) are a major public health problem—with over 200 million people reporting drug use in 2016. Electroencephalography (EEG) is a powerful tool that can provide insights into the impact of SUDs on cognition. Specifically, modulated gamma activity may provide an index of the pathophysiology of SUDs. Thus, the purpose of this review was to investigate the impact of alcohol, tobacco, cannabis, cocaine, and amphetamine on gamma activity, among pre-clinical and clinical populations during acute and chronic exposure and withdrawal states. We searched multiple databases for key terms related to SUDs, EEG, and gamma and ensured rigorous methods by using a standardized review reporting tool. We included 30 studies in this review and found that all substances were associated with modulation of gamma activity, across states and in both preclinical and clinical populations. Gamma oscillations appeared to be differentially modulated in clinical versus preclinical populations and had the most complex relationship with alcohol, indicating that it may act differently than other substances. The findings of this review offer insights into the pathophysiology of SUDs, providing a potential window into novel treatments for SUDs via modulation of gamma activity.

## 1. Introduction

Substance abuse represents a rapidly growing global public health concern. According to the United Nations Office on Drugs and Crime, 271 million people used at least one illicit drug in 2016 [1], of which 35 million have developed substance use disorders (SUDs) [1]. Approximately 2.3 billion people self-reported as current drinkers in 2016 [2], more than 1 billion people around the world smoked tobacco in 2019 [3] and 188 million people used cannabis in 2017 [1]. Furthermore, 35 million people worldwide have SUDs but less than 15% receive treatment [1].

Cognition has been shown to be modulated with acute and chronic substance use. For example, studies have found that working memory, the ability to maintain and manipulate information over short periods of time [4], has been impaired with alcohol and cannabis use disorders (AUDs and CUDs) [5,6]. Chronic alcohol use may cause pathological changes in the brain, such as ventricle enlargement [7] and reductions in grey and white matter volume [8,9], which have been linked to cognitive impairment [8]. Cognitive deficits have also been reported in individuals following acute alcohol intoxication, including disruptions in memory [10], divided attention [11], and information processing [12]. In addition, chronic non-psychiatric cannabis and cocaine users have demonstrated impaired learning [13,14], memory [14,15,16,17], attention [13,15,16,18], and executive function [19]. Interestingly, cognition has been shown to rebound with discontinuing substance use (e.g., cannabis [20]), thereby furthering support for the association between cognition and SUDs, possibly through shared pathophysiology [21]. Investigating different states (e.g., acute, chronic, and withdrawal) of substance use on such pathophysiology may help to optimize the treatment for SUDs and associated cognitive deficits. 

Electroencephalography (EEG) is a powerful non-invasive neurophysiological imaging technique with high temporal resolution that can measure cortical neural activity through recording from scalp electrodes [22]. There are five conventional EEG frequency bands—delta (1–3.5 Hz), theta (4–7 Hz), alpha (8–12 Hz), beta (12–28 Hz), and gamma (30–120 Hz). Gamma oscillatory activity has been associated with higher-order perceptual and cognitive processes, including long-term memory [23], selective attention [24,25], visual search [26,27], learning [28], and working memory [29,30,31]. In fact, increases in gamma power have been associated with increases in working memory load in healthy subjects [29,30,32] and epileptic patients [31]. Patients with schizophrenia, however, lack this ability to modulate gamma activity [30,32]. Gamma oscillations therefore may provide a window into ongoing sensory-cognitive processes in the brain.

Gamma activity has been recorded in multiple brain regions, including the hippocampus [33], neocortex [34,35,36,37], entorhinal cortex [38], olfactory bulb [39], and amygdala [40]. Gamma-aminobutyric acid (GABA) is an inhibitory neurotransmitter present in large quantities in the mammalian brain. Networks of specialized GABAergic interneurons (in particular, basket and chandelier cells; [41]) and pyramidal cells are responsible for generating a large-scale gamma oscillatory activity [42,43]. Two types of GABAergic receptors—GABA_A_ and GABA_B_ receptors—are linked to the generation [43,44,45] and modulation [43] of gamma oscillations, respectively. Gamma activity can also be produced through activation of several other receptors [45], including the metabotropic glutamate receptor [43], the muscarinic-cholinergic receptor [46], and the kainate receptor [47]. The N-methyl-D-aspartic acid (NMDA) receptor has also been shown to influence gamma activity [48]. Given that gamma oscillations underlie several cognitive domains and are altered by substance use, deficits in gamma band activity may be a valuable index of the pathophysiology of SUDs.

The goal of this paper is to provide a comprehensive review of the existing body of literature on the effects of alcohol, tobacco, cannabis, cocaine, and amphetamine use on gamma oscillatory activity, among preclinical and clinical populations during acute, chronic, and withdrawal states.

## 2. Methods

A comprehensive literature search was conducted using the databases PubMed, MEDLINE, and PsycINFO on 3 September 2019 with no date restrictions. Search terms included “cannabi*”, “marijuana”, “THC” (tetra-hydrocannabinol), “delta-9-tetra-hydrocannabinol”, “alcohol”, “ethanol”, “EtOH”, “nicotine”, “tobacco”, “cigarette”, “cocaine”, “amphetamine”, and “methamphetamine” with “EEG”, “electroencephalography”, “gamma oscillat*”, “gamma activity”, or “gamma power”. Both relevant preclinical and clinical studies published in English were included. Studies were excluded if they investigated populations with psychiatric disorders other than alcohol, cannabis, tobacco, amphetamine, or cocaine use disorder; or schizophrenia. Studies were excluded if publication types were non-empirical research studies, editorials, opinion articles, protocols, abstracts, proceedings, conceptual analyses, case studies, patient resources, or reviews. Reference lists of reviews were manually searched for additional studies. The search produced 524 studies, and a total of 30 articles were included in the review. We complied with the Preferred Reporting Items for Systematic Reviews and Meta-Analyses (PRISMA) guideline, to ensure rigour (Figure 1) [49]. Given that oscillations in the gamma frequency are tightly linked to cognition, and that cognition is altered following substance use, gamma oscillations serves as a useful neurophysiological index for examining the underlying pathophysiology of SUDs for the development of novel treatment options.

## 3. Results

### 3.1. Preclinical Studies

Alcohol: There are a limited number of studies that examined gamma oscillatory activity following acute ethanol exposure (Table 1). Wang et al. reported a dose-dependent relationship between acute ethanol exposure and gamma activity in the rat hippocampal CA3 area [50]. At a low dose (10 mM), a minor increase in gamma power was observed. However, at higher doses (25–100 mM), alcohol significantly suppressed gamma activity, with the greatest reduction occurring at 100 mM (52.9 ± 8.5%). Peak gamma frequency was not altered with exposure [50]. In another study, Tsurugizawa and colleagues investigated the effects of alcohol on local field potential (LFP) oscillations in the nucleus accumbens of 34 male Wistar rats [51]. The animals were injected intraperitoneally with 0.4 g/kg of ethanol solution. Following ethanol exposure, alpha and beta power decreased while gamma power increased significantly [51].

Several studies to date have examined the effects of ethanol withdrawal on the EEG spectral profiles of mice and rats (Table 1). Ehlers and Chaplin found significant elevations in cortical EEG power across almost all frequency bands, including gamma, 24 h following termination of prolonged ethanol exposure [52]. This was also observed in a study by Slawecki et al., investigating the withdrawal responses of male adolescent and adult rats [53]. Following ethanol vapour exposure for 14 consecutive days, prepulse inhibition increased in both groups. Although changes in frontal gamma power were insignificant, beta and gamma power in the parietal cortex significantly increased in adolescent rats during withdrawal [53]. In addition to increases in parietal gamma power, Cheaha and colleagues also found enhanced gamma activity in the frontal region, as well as increased locomotor activity and decreased sleep time in mice experiencing ethanol withdrawal [54]. These preclinical studies demonstrate that ethanol withdrawal is associated with increased gamma band activity, indicating central nervous system hyperexcitability during withdrawal states.

Tobacco: Several animal studies have reported dose-dependent relationships between acute nicotine exposure and gamma band activity (Table 2). Phillips and colleagues administered nicotine (1.0 mg/kg), mecamylamine (2.0 mg/kg), saline, or nicotine combined with mecamylamine in 11 male mice [55]. It was found that nicotine significantly increased auditory evoked gamma power. Moreover, pretreatment with the nicotinic antagonist mecamylamine prevented this nicotine-induced increase [55]. Song et al. also reported an enhancement in gamma activity in the rat hippocampal slices following nicotine infusion under high doses [56]. In this study, administering 10 µM of nicotine did not significantly influence tetanically-induced gamma oscillations under both threshold stimulation intensity and double threshold stimulation intensity [56]. However, nicotine infusion at 100 µM significantly increased both tetanic gamma power and frequency [56]. Interestingly, a recent study found a significant increase in the amplitude of kainate-induced gamma activity after just 1 µM of nicotine application [57]. Furthermore, in a study conducted by Wang et al., nicotine at lower doses (0.1–10 µM) increased gamma power and slightly reduced peak gamma frequency [58]. The greatest increase in gamma power (83 ± 21%) was also achieved after 1 µM of nicotine application. However, contrary to Song et al.’s findings, Wang and colleagues reported that nicotine decreased gamma power at 100 µM [56,58].

One study explored the effects of acute nicotine exposure (100 µM) on the complexity of gamma oscillations in the mouse hippocampus ([59] Table 2). It was found that nicotine reduced gamma oscillation complexity compared to the control and washout conditions, indicating increased synchronization of hippocampal networks.

Recently, Bueno-Junior and colleagues examined the effects of acute versus chronic nicotine exposure on brain oscillatory activity in rats [60]. The authors implemented a daily dosing regimen of nicotine (0.2 mg/kg) in vivo and found that acute nicotine administration slightly increased gamma power and reduced theta and beta power. Daily nicotine exposure produced stronger and more robust gamma activity [60]. In the following experiment, a separate group of rats was treated with nicotine following the same daily dosing regimen. Nicotine-treated rats performed significantly better in a visual attention task compared to controls, verifying the cognitive enhancing effects of nicotine [60]. In sum, nicotine administration increases gamma band activity. According to most preclinical studies, this increase in gamma power can be achieved with doses as low as 0.1 µM.

Cannabis: Studies have shown that chronic exposure to THC-9 in adolescent rodents induces long-lasting deficits in neural oscillations during adulthood (Table 3). For example, Raver and colleagues [61,62] showed that repeated exposure to THC in adolescents, but not adults, suppresses pharmacologically evoked cortical oscillations and impairs working memory performance in adulthood. In vitro recording of local field potentials show reduced oscillatory activity across all frequencies, whereas in vivo electrocorticogram measured specific reductions in alpha and gamma oscillations. The areas with the most pronounced deficits in gamma oscillatory activity in adulthood are also regions that were least mature during THC exposure. These preclinical findings provide insight into how adolescent cannabis use may affect gamma oscillations in adulthood, potentially as a precursor of CUDs. Overall, preclinical studies demonstrate that cannabis exposure is associated with reduced gamma activity.

Cocaine: Studies examining the effects of cocaine use on gamma oscillation are limited (Table 4). In a preclinical acute administration study, Dilgen and colleagues examined the effects of acute cocaine exposure on prefrontal cortex optogenetically evoked gamma oscillation in PV-Cre knock-in mice [63]. The results showed that acute cocaine administration resulted in a decrease in the spread of induced gamma oscillations, likely due to more synchronous principal neuron firing. The authors suggest that the increase in synchrony coupled with an increase in the entrainment of gamma oscillations may be a potential mechanism in which cocaine increases alertness in novice users.

Amphetamines: Studies investigating the role of amphetamines in neural oscillations are limited to rodent populations (Table 5). Janetsian and colleagues induced methamphetamine sensitization in rats as assessed temporal and recognition memory after 1 or 30 days of abstinence as well as recorded oscillatory activity in the medial prefrontal cortex [64]. Results showed that temporal memory was impaired after both 1 day and 30 days abstinence, while recognition memory was only impaired after 1 day of abstinence. Methamphetamine injections significantly decreased neuronal firing rates and pharmacologically induced slow gamma oscillations in both sensitized and control rats. Moreover, the number of neurons phase-locked gamma oscillations were reduced in methamphetamine sensitized rats compared to controls. Given that previous studies have demonstrated that gamma band activity is elevated during recognition tasks in both humans and rats, it has been suggested that deficits in memory following repeated methamphetamine exposure may be a result of altered gamma band activity [64]. A study by Morra and colleagues investigated the role of cannabinoid (CB) 1 receptor in methamphetamine-induced rodent stereotypies and disrupted gamma oscillations [65]. Prior to methamphetamine sensitization, rats were treated with either CB1 receptor antagonist rimonabant (0.3 mg/kg) or vehicle as control. Methamphetamine administration in control rats significantly increased high-frequency gamma oscillation power (70–94 Hz, ~80 Hz peak) in the nucleus accumbens. Interestingly, the methamphetamine mediated increase in high-frequency gamma oscillations was attenuated by CB1 receptor blockades in rimonabant treated rats. This finding suggests that CB1 receptor activity is implicated in the mechanism in which methamphetamines alter local gamma oscillations. Furthermore, motor stereotypy behaviours induced by methamphetamine were also diminished in CB1 antagonist treated rats, in line with previous findings that both CB1 KO mice and local CB1 receptor antagonist injections in Nucleus accumbens reduce amphetamine-induced stereotypies. Taken together, these studies provide evidence that acute methamphetamine administration disrupts normal gamma oscillation and that its ability to do so may be modulated by CB1 activity.

### 3.2. Clinical Studies

Alcohol: Two studies have examined gamma band activity following acute alcohol consumption in human subjects (Table 6). In an early study conducted by Jaaskelainen and colleagues, alcohol (10% (*v*/*v*)) administered at moderate doses (0.50 g/kg and 0.75 g/kg) significantly reduced gamma power [66]. In this randomized and double-blind study, 10 healthy young adults with a negative family history for alcoholism participated in a dichotic listening task after consuming 0.25, 0.50, or 0.75 g/kg of 10% alcohol beverage or placebo. The effect of alcohol on gamma power was insignificant at 0.25 g/kg, but doses 0.50 and 0.75 g/kg markedly suppressed gamma power in both the attended and non-attended task conditions, suggesting that blood alcohol concentrations of around 0.05% already begin to influence gamma activity in healthy humans [66]. However, Campbell et al. demonstrated that alcohol administered at a higher concentration (40% (*v*/*v*)) and dose (0.8 g/kg) increased peak gamma power in both the human visual and motor cortices and decreased peak gamma frequency in the visual cortex [67].

Padmanabhapillai and colleagues investigated early evoked gamma oscillations in 122 alcohol-dependent individuals and 72 healthy controls during a visual oddball task [68]. In this study, alcoholic subjects reported an average of 10.07 drinks per drinking day. Controls were social drinkers with a mean of 1.68 drinks per drinking day. Gamma activity was found to be significantly lower in alcoholics during target stimuli processing compared to controls but significantly higher during non-target processing [68]. In addition to evoked gamma power, reductions in evoked delta and theta band activity during stimulus processing have also been previously reported in patients with chronic alcoholism [69]. Following a similar experimental design, the same group then investigated early evoked gamma oscillations in male adolescents at high risk for alcoholism [68]. Consistent with alcohol-dependent individuals, high-risk adolescents also exhibited lower evoked gamma power during target processing in both the frontal and parietal regions compared to controls [68]. Furthermore, while control subjects displayed higher evoked gamma band responses in target-present conditions compared to non-target and novel-stimuli conditions, high-risk subjects did not display this pattern [68]. Moreover, De Bruin and colleagues demonstrated that heavy alcohol drinkers may also undergo changes in hippocampal-neocortical connectivity [70]. In this study, 11 male heavy drinkers and 11 male light drinkers participated in two EEG recording sessions. Heavy drinkers were defined as consuming over 360 g of alcohol per week whereas light drinkers consumed less than 360 g per week. Heavy drinkers exhibited higher theta and gamma synchronization compared to light drinkers in both conditions [70].

To summarize, the relationship between alcohol and gamma band activity appears to be a dose-dependent and inverted U-shaped pattern—administration at a low dosage generally increases gamma power while gamma activity is suppressed by alcohol at intoxicating concentrations. In addition, chronic alcohol consumption may lead to reduced gamma power as well as altered brain connectivity, which may underlie impaired cognitive function observed with severe AUDs.

Tobacco: Gamma activity following acute nicotine exposure has also been studied in human tobacco smokers (Table 7). Crawford and colleagues examined sensory gating, P50, and auditory evoked gamma oscillations, among sex- and age-matched cigarette smokers and never-smokers [71]. In this study, smokers had smoked for more than 5 years and currently smoke at least 20 cigarettes per day. Both groups of participants were tested in two conditions—smokers tested after overnight abstinence and after smoking, while controls did not smoke. Sensory gating refers to the process of filtering stimuli, by separating irrelevant stimuli from meaningful stimuli [72]. Crawford et al. examined sensory gating using a paired-tone paradigm, where two identical auditory tones are played 50-milliseconds apart, and participant sensory gating levels are measured by calculating the ratio of P50 event-related potential amplitudes [71]. They found that smokers had significantly greater gamma power compared to never-smokers and also displayed greater sensory gating both before and after abstinence [71].

No study to date has yet examined changes in gamma band activity in human subjects following a prolonged nicotine administration paradigm. However, a study by Wilbanks et al. did examine the impact of tobacco smoking on resting neural oscillations in postpartum mothers [73]. Although delta, theta, and alpha power were found to be elevated in smokers, no significant difference in gamma oscillatory power was observed in smoking mothers compared to controls [73] (Table 7).

Generally, nicotine administration increases gamma band activity. In chronic tobacco smokers—compared to never-smokers, smokers have higher gamma power. The above findings support the hypothesis that nicotine may help improve neurocognitive dysfunction in individuals with schizophrenia [74] through the modulation of gamma oscillations [75].

Cannabis: Studies of acute THC administration in human populations suggest that cannabis disrupts gamma oscillatory activity (Table 8). Nottage and colleagues investigated the effects of intravenous THC on high-frequency EEG recordings compared to placebo in participants with prior exposure to cannabis use but who were not cannabis-dependent [76]. THC shifted oscillatory activities towards higher frequencies such that there was a decrease in high beta oscillation (21–27 Hz) and increased low gamma (27–45 Hz) at rest. Furthermore, reductions in the high beta range (21–27 Hz) were more prominent in anterior regions, while posterior regions generally showed increases in the low gamma range (27–45 Hz). The THC-induced shift to faster gamma oscillations resulting in a hyperactive cortex may be associated with saliency misattribution in delusional states observed with psychosis.

Another study by Cortes-Briones and colleagues also demonstrated the association between THC and effects on the positive and negative symptom scale (PANSS)—an index used among patients with schizophrenia. In this study, an auditory steady-state response paradigm to probe the effects of acute THC administration on broadband-frequency neural oscillations and its relation to psychosis-related effects in healthy, cannabis naïve, and recreational users [77]. Auditory steady-state evoked potentials were assessed in both cannabis using and cannabis naïve subjects at varying auditory stimulation frequencies. The study demonstrated for the first time that acute THC administration induces a dose-dependent reduction of intertrial coherence (ITC) at 40 Hz stimulation frequency, suggesting that exogenous THC disrupts time-locked gamma band activity. Moreover, there was an inverse relationship between ITC and positive symptoms as measured by the PANSS, implicating gamma activity in positive symptom presentation among a healthy population. These findings from acute administration trials thus provide evidence that cannabis use disrupts gamma band activity among the general population. Moreover, the effect of THC on gamma oscillations may represent the pathophysiological mechanism through which psychosis is mediated.

Clinical studies examining chronic cannabis use and its effects on EEG recordings provide further evidence that cannabinoid exposure contributes to positive symptoms in psychosis by disrupting gamma band activity. Researchers used the auditory steady-state response paradigm to examine the effects of chronic cannabis use on broadband-frequency neural oscillations in cannabis using and cannabis naïve controls [78]. Cannabis users showed reduced evoked 40 Hz harmonic during 20 Hz stimulation frequencies compared to controls. Evoked power values at 20 Hz stimulation frequency were also negatively correlated with scores of schizotypal personality characteristics. Cannabis using subjects demonstrated increased schizotypal characteristics compared to cannabis naïve subjects as assessed by the Schizotypal Personality Questionnaire. Total years of cannabis use were also positively correlated with higher measures of schizotypal characteristics. In a follow-up study, Skosnik again used the auditory steady-state response paradigm to investigate disruptions in broadband EEG neural oscillation associated with cannabis use in a larger sample with more variations in click-train frequencies [79]. Results showed a selective and significant decrease in Fourier-based mean trial power (MTP) at 40 Hz among cannabis users compared to cannabis naïve controls, while an earlier onset of cannabis use was associated with lower oscillatory power at 40 Hz. Edwards and colleagues investigated the effects of chronic cannabis use on auditory evoked neural oscillation and auditory P50 sensory gating [80]. Results from event-related spectral perturbations (ERSP) analysis showed abnormal high-frequency activity in gamma range (30–50Hz) in cannabis users following auditory stimulation. Cannabis users demonstrated a time-specific reduction in power in high-frequency gamma and beta bands following the first stimulus (S1) and reductions in just gamma power following the second stimulus (S2). P50 gating was reduced in the cannabis using population compared to cannabis naïve controls in a manner that is similar to gating disturbances found among patients with schizophrenia. Additionally, greater levels of cannabis use were associated with high P50 ratios and negatively correlated with ERSP gamma power [80]. In a study assessing gamma oscillation during a coherent motion task in chronic cannabis users and cannabis naïve controls, cannabis users demonstrated a significant decrease in induced gamma power in coherent conditions compared to controls [81]. Moreover, an increasing trend in the Perceptual Aberration Scale (PAS), with higher scores positively correlated with total years of cannabis use, was observed among chronic cannabis users [81] (Table 8).

Overall, these studies suggest that cannabis, and particularly THC, decreases gamma activity. Clinical studies indicate that the disruption of gamma activity resulting from both acute THC administration and chronic cannabis use is associated with central nervous system hyperexcitability and positive symptomatology typically observed in psychosis.

Cocaine: One human study examined attentional bias to drug related cues by measuring evoked and induced gamma reactivity measure in patients with cocaine use disorder before and following motivational interview-based neurofeedback treatment [82] (Table 9). It is interesting to note that out of 10 participants, 7 also tested positive for cannabis use. Following 12 sessions of treatment, cocaine use decreased marginally while cannabis use significantly decreased for all patients. EEG reactivity to drug related cues was decreased post-treatment and, specifically, both evoked and induced gamma power decreased globally for drug related cues and non-drug related cues. While it is difficult to correlate the decreased cocaine usage to changes in neurophysiology given the nature of the self-reporting method, the accidental finding of dramatic decreases in cannabis use further cofounds the investigation into cocaine effects on gamma oscillations.

Only two studies reported on cocaine and gamma activity (Table 9). Dilgen et al. found a decrease in gamma activity following acute administration of cocaine in a preclinical population [63]. Horrell et al. found decreased gamma activity among chronic cocaine users, persisting post-treatment [82]. These studies suggest that cocaine use may result in long-term decreases in gamma activity.

## 4. Discussion

In this paper, we provide a comprehensive review of current evidence on alcohol, tobacco, cannabis, cocaine, and amphetamine use on gamma oscillatory activity during acute, chronic, and withdrawal states, among preclinical and clinical populations. All substances were associated with modulated gamma activity, in both preclinical and clinical populations and across states. Moreover, gamma appeared to be differentially modulated by acute versus chronic exposure, which may be related to cognitive dysfunction associated with substance abuse.

Pre-clinical studies indicated that all five substances modulated gamma activity. Alcohol use displayed an inverted U-shaped pattern during exposure for gamma and an increase across all frequencies during withdrawal. Tobacco use resulted in increased gamma during both acute and chronic exposure. Acute cannabis and cocaine exposure resulted in a decrease in gamma. Amphetamine use had mixed effects on gamma activity.

Similarly, clinical studies demonstrated that alcohol, tobacco, and cannabis modulated gamma activity. Acute alcohol exposure resulted in a selective decrease in gamma among non-dependent individuals, while such decreases were observed across all frequency bands among dependent individuals. Chronic tobacco use led to an increase in gamma, whereas chronic cannabis use led to a decrease in gamma. Clinical studies of cocaine and amphetamine were limited, making it difficult to draw any patterns on their influence on gamma. Thus, it appears that alcohol may act differently than other substances.

Alcohol and gamma had the most complex relationship, with alcohol decreasing gamma during withdrawal and in non-dependent individuals but displaying an inverted U-shaped pattern during exposure. Findings were mixed, likely to do studies varying in tasks used, types of controls (e.g., social drinkers, light drinkers, never drinkers), participant state (i.e., exposure versus withdrawal), and whether alcohol exposure was acute or chronic. The preclinical evidence of increased gamma activity during periods of alcohol withdrawal may represent post alcohol withdrawal hyperexcitability, a phenomenon whereby alcohol inhibits NMDAR excitatory signalling and increases GABAR inhibitory signalling, leading to central nervous system suppression and resulting in hyperexcitability during withdrawal [83,84]. The clinical evidence of varying effects of alcohol on gamma based on the task used suggests that there may be differential effects of alcohol based on the brain area involved and the type of cognition involved, a finding previously noted in the literature [85,86,87]. These findings indicate that several brain areas may be involved in the neuropathology of AUDs.

This review contains several strengths and limitations. We searched multiple databases and followed the PRISMA reporting guidelines to ensure a rigorous standard of review and reporting. We also searched for a variety of terms in order to capture all potentially relevant studies. A limitation of this review was that we only reviewed the five most common SUDs (alcohol, tobacco, cannabis, cocaine, and amphetamine), potentially limiting the generalizability of findings to other substances. Next, we did not directly search for the term “schizophrenia” or “psychosis”; we instead focused on search terms related to gamma, EEG, and SUDs and then manually screened for studies including patients with schizophrenia. This may have led to unintended exclusion of relevant studies. Last, we excluded studies of gamma in other disease areas, such as the effects of gamma entrainment therapies on brain pathology and cognitive symptoms in Alzheimer’s disease (e.g., [87,88,89]). This may have omitted potentially relevant studies; however, we intentionally focused on SUDs and schizophrenia, given the prevalence and lack of research on these disorders.

## 5. Conclusions and Implications

The findings of this review indicate that both acute and chronic substance use modulate gamma activity and do so across different states. These findings offer insights into the potential mechanism underlying the pathophysiology of SUDs. This review also contributes to the growing body of evidence indicating the potential for brain stimulation to address deficits associated with SUDs. Brain stimulation treatments, such as transcranial direct current stimulation and repetitive transcranial magnetic stimulation, that modulate gamma activity may offer a promising potential approach for targeting working memory. Further research is needed to fully understand the underlying pathophysiology of SUDs in order to advance treatment of these devastating disorders and reduce the global burden on healthcare systems.

## Figures and Tables

**Figure 1 jpm-11-00017-f001:**
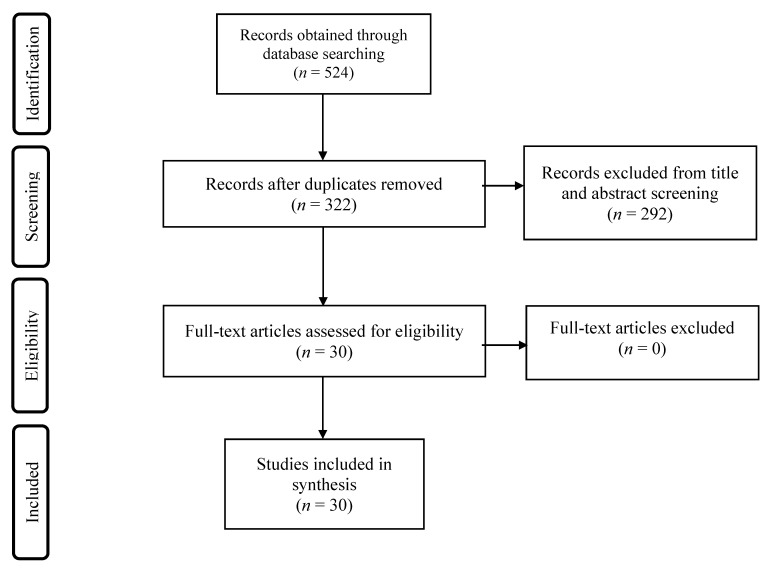
PRISMA flow diagram of the screening process.

**Table 1 jpm-11-00017-t001:** Preclinical Studies Examining the Effects of Alcohol on Gamma Oscillations.

Study	Gamma Frequency Range and Type	Subject Information	Method of Administration	Main Findings
Ehlers and Chaplin (1991)	32–64 Hz	38 Wistar rats	28-day continuous exposure to ethanol vapour (22–28 mg/L O_2_)	Increased power in all frequency bands except 1–2 Hz and 16–32 Hz in the cortex 24 h after ethanol exposure. No differences in gamma power in both the cortex and dorsal hippocampus 10–30 min and 2 weeks following ethanol exposure.
Slawecki et al. (2006)	32–50 Hz	136 Male Sprague-Dawley rats (adolescents: 28–30 days; adults: 60–70 days)	14-day (12 h/day) continuous exposure to ethanol vapour (95% ethanol)	Increased parietal gamma and beta (16–32 Hz) power in adolescent rats during acute ethanol withdrawal (7–10 h post-exposure) on days 8 and 12. No change in frontal gamma power during withdrawal in both adolescent and adult rats.
Cheaha et al. (2013)	30.5–45 Hz	Male Wistar rats	Ethanol-containing modified liquid diet at steadily increasing concentrations for 28 days	Increased frontal and parietal gamma power during acute ethanol withdrawal (1–8 h after ethanol exposure).
Campbell et al. (2014)	30–80 Hz; Induced	16 Social drinkers (8 males, 8 females; mean age: 25.9 years) with average weekly alcohol consumption of 191.2 g for males and 132 g for females	40% (*v*/*v*) alcohol solution or placebo (males consumed 0.8 g/kg, females consumed 0.72 g/kg)	Increased peak gamma power in the visual and motor cortices. Decreased peak gamma frequency in the visual cortex. No change in peak gamma frequency in the motor cortex.
Tsurugizawa et al. (2016)	60–80 Hz; Induced	34 Male, alcohol-naïve Wistar rats (8–12 weeks)	Intraperitoneal injection of 0.4 g/kg of ethanol solution	Increased gamma power that peaks within 5 min after ethanol injection and returns to baseline at around 10 min post-injection.
Wang et al. (2016)	30–80 Hz; Induced (kainate)	Male Sprague Dawley rats (4–5 weeks)	5–100 mM ethanol added to artificial cerebrospinal fluid	No change in gamma power after 5 mM ethanol application. Increased gamma power after 10 mM ethanol application. Decreased gamma power following 25–100 mM ethanol application (dose-dependent). No change in peak gamma frequency following ethanol exposure.

**Table 2 jpm-11-00017-t002:** Preclinical Studies Examining the Effects of Tobacco on Gamma Oscillations.

Study	Gamma Frequency Range and Type	Subject Information	Method of Administration	Main Findings
Song et al. (2005)	30–80 Hz; Induced (tetanic stimulation)	Wistar rats (17–30 days)	(-) Nicotine dissolved in artificial cerebrospinal fluid	- No change in gamma power after 10 µM of nicotine perfusion at both threshold and double threshold stimulation intensity. - Increased gamma power and frequency after nicotine perfusion of 100 µM at both stimulation intensities.
Phillips et al. (2007)	31–61 Hz; Evoked (auditory)	11 Male C57BL/6 J mice (10–12 weeks)	Intraperitoneal injection of 0.1 mL of nicotine hydrogen tartrate salt (1.0 mg/kg) dissolved in saline (0.09%)	- Increased gamma power following nicotine administration.
Akkurt et al. (2010)	30–80 Hz; Induced (auditory)	Sprague-Dawley rats (23–36 days)	100 µM (-) Nicotine dissolved in artificial cerebrospinal fluid	- Decreased gamma oscillation complexity during nicotine exposure. - Decreased gamma oscillation complexity during nicotine washout.
Zhang et al. (2015)	20–80 Hz; Induced (kainate)	Male Wistar rats (3 weeks)	Nicotine sulfate	- Increased gamma power following 1 µM of nicotine administration versus control. - Increased gamma power during nicotine washout versus control.
Wang et al. (2015)	20–60 Hz; Induced (kainate)	Male Wistar rats (4–5 weeks)	0.1–100 µM nicotine sulfate dissolved in artificial cerebrospinal fluid	- Increased gamma power following 0.1–10 µM of nicotine administration. - Decreased peak gamma frequency following nicotine administration at concentrations 0.25–10 µM. - Decreased gamma power and no change in peak gamma frequency after 100 µM of nicotine administration.
Bueno-Junior et al. (2017)	40–130 Hz (low gamma: 40–60 Hz; high gamma: 60–130 Hz)	Male Long-Evans rats (85–90 days)	Daily intraperitoneal injection of 0.2 mg/kg of nicotine solution (dissolved nicotine hydrogen tartrate salt in saline) on days 1–5, followed by a 9-day washout, and a final nicotine injection after washout.	- Increased low gamma power following nicotine administration on days 1, 3, 5, and 15. - No change in high gamma power following nicotine exposure on day 1 but increased high gamma power on day 15.

**Table 3 jpm-11-00017-t003:** Preclinical Studies Examining the Effects of Cannabis on Gamma Oscillations.

Study	Gamma Frequency Range and Type	Subject Information	Method of Administration	Main Findings
Raver et al. 2013	Gamma 30–80 Hz (Local Field Potentials and Power)	Male CD-1 Mice	THC (5 mg/kg) dissolved in 100% ethanol and injected in a 1:1:18 solution of ethanol castor oil: 0.9% saline (1 mL/kg)	Cannabis exposure suppresses evoked cortical oscillations (with marked reductions in gamma and alpha) and impairs working memory in adolescent but not adult mice.
Raver et al. 2014	Gamma 30–80 Hz (Local Field Potentials and Power)	Male CD-1 Mice	CB1R/CBR2 agonist WIN55, 212-2 (1 or 2 mg/kg), CB1R inverse agonist/antagonist AM251 (0.3, 0.5, 1 or 2 mg/kg), CB1R/CB2R agonist (THC 5 mg/kg) and putative CB1R-inactive enantiomer (WIN55, 212-3) dissolved in 100% ethanol and administered in a 1:1:18 solution of ethanol: 0.9% saline at final volume of 1 mL/kg	THC selectively suppresses oscillations in the medial prefrontal cortex mediated by CB1R and non-cannabinoid receptors.

**Table 4 jpm-11-00017-t004:** Preclinical Studies Examining the Effects of Cocaine on Gamma Oscillations.

Study	Gamma Frequency Range and Type	Subject Information	Method of Administration	Main Findings
Dilgen et al. 2013	Relative and peak power (1–100 Hz)	Male PV-Cre Mice (B6; 129P2-Pvalb)	Infusion (cocaine HCL, SCH 23390 15 mg/kg)	Acute cocaine administration increased the entrainment of gamma oscillations to the optogentically induced driving frequency.

**Table 5 jpm-11-00017-t005:** Preclinical Studies Examining the Effects of Amphetamine on Gamma Oscillations.

Study	Gamma Frequency Range and Type	Subject Information	Method of Administration	Main Findings
Janetsian et al. 2015	Gamma power (30–50 Hz)	Male adult Sprague-Dawley rats	5.0 mg/kg of methamphetamine	Temporal memory was impaired after 1 and 30 days of abstinence. Injection of MA decreased neuronal firing rate and anesthesia-induced slow oscillation in both sensitized and control rats. Relationships were found between anesthesia-induced slow oscillation and gamma power. Decreased number of neurons phase-locked gamma frequency was observed in the sensitized rats.
Morra et al. 2012	Oscillatory Power 0–100 Hz)	Male adult Sprague-Dawley rats *n* = 10	Intravenous CB1 receptor antagonist rimonabant (0.3 mg/kg) or vehicle followed by an ascending dose regimen of methamphetamine (0.01, 0.1, 1 and 3 mg/kg)	Methamphetamine increased high frequency gamma oscillations (~80 Hz). Methamphetamine induced both stereotypy and high frequency gamma power that was later disrupted with CB1R blockade.

**Table 6 jpm-11-00017-t006:** Clinical Studies of Alcohol and Gamma Oscillations.

Study	Gamma Frequency Range and Type	Subject Information	Method of Administration	Main Findings
Jaaskelainen et al. (2000)	40-Hz; Evoked	10 Social drinkers FHNFA * (5 males and 5 females; age: 20–28 years)	0.25, 0.50, or 0.75 g/kg of 10% (*v*/*v*) ethanol solution or placebo	No differences in gamma power after consuming 0.25 g/kg. Decreased gamma power following ingestion of the 0.50 g/kg and 0.75 g/kg doses.
De Bruin et al. (2004)	30–45 Hz	22 Male social drinkers FHNFA (11 light drinkers: <360 g alcohol per week; 11 heavy drinkers: >360 g alcohol per week; age: 22–27 years)	---	Increased gamma and theta (4–8 Hz) synchronization in heavy drinkers compared to light drinkers. Insignificant between-group differences in relative gamma and theta power.
Padmanabhapillai et al. (2006a)	29–45 Hz; Evoked	122 people with alcoholism (male; age: 20–40 years), 72 social drinkers FHNFA (male; age: 19–36 years)	---	Decreased frontal gamma power in alcoholics compared to controls during target processing. Increased frontal gamma power in alcoholics compared to controls during non-target processing.
Padmanabhapillai et al. (2006b)	29–45 Hz; Evoked	68 Male adolescents with at least one alcohol-dependent parent (high-risk), 27 male adolescents from non-alcoholic families (low-risk)	---	Decreased frontal and parietal gamma band response in high-risk adolescents compared to controls during target processing. No change in gamma activity between target, non-target and novel stimuli conditions in the high-risk group.

* FHNFA: family-history negative for alcoholism.

**Table 7 jpm-11-00017-t007:** Clinical Studies of Tobacco on Gamma Oscillations.

Study	Gamma Frequency Range and Type	Subject Information	Method of Administration	Main Findings
Crawford et al. (2002)	32–48 Hz; Evoked	13 Heavy cigarette smokers (>20 cigarettes per day) and 13 age- and sex-matched never-smokers (age: 20–40 years)	Smokers were assessed following overnight (9–15 h) abstinence and after smoking their usual brand of cigarettes	- Increased gamma power in smokers versus controls.
Wilbanks et al. (2016)	30–80 Hz	35 Smokers and 35 age- and demographically matched never-smokers three months postpartum	---	- No change in gamma, high alpha (10.5–13 Hz), and beta (13–30 Hz) power between smoking and non-smoking mothers. - Increased delta (1–4 Hz), theta (4–8 Hz) and low alpha (8–10.5 Hz) power in smokers compared to controls.

**Table 8 jpm-11-00017-t008:** Clinical Studies of Cannabis on Gamma Oscillations.

Study	Gamma Frequency Range and Type	Subject Information	Method of Administration	Main Findings
Nottage et al. 2015	Resting state low gamma (35–45 Hz ad event-related synchronization (ERS) during motor associated high gamma (65–85 Hz)	14 Human Subjects	Intravenous THC (1.25 mg)	THC induced a shift to faster gamma oscillations and may represent an over-activation of the cortex that was related to positive symptoms.
Cortes-Briones et al. 2015	Auditory steady-state at 20, 30, and 40 Hz evoked potentials (Inter-trial coherence and evoked power ~40 Hz)	Human Subjects *n* = 20	Intravenous THC (0.003 mg/kg)	THC reduced ITC in the 40 Hz condition and evoked gamma power compared to placebo Negative correlation was observed between 40 Hz ITC and PANSS subscales
Skosnik et al. 2006	Auditory steady-state evoked potentials (spectral power) during auditory click trains of 20, 30, and 40 Hz)	Human Subjects (Current cannabis users *n* = 17 and drug naïve *n* = 16)	---	Reduced power during the 20 Hz stimulation frequency among cannabis users that were correlated with schizotypal personality questionnaire scores.
Skosik et al. 2012	Auditory steady-state evoked potentials (spectral power) during auditory click trains at 9 different frequencies)	Human Subjects (Chronic cannabis users *n* = 22 and cannabis naïve controls *n* = 24)	---	Decreased spectral power was observed among cannabis users. Reduced gamma power was related to an earlier age of onset of cannabis use. No effects on phase-locking or the N100, suggesting that cannabis may selectively impair the ability to generate oscillations in the gamma frequency range.
Edwards et al. 2009	Gamma range (30–50 Hz) during event-related spectral perturbations (ERSP) and inter-trial coherence (ITC)	Human Subjects (Heavy cannabis users *n* = 17 and cannabis naïve *n* = 16)	---	Reduced P50 gating and attenuated ITC among heavy cannabis users compared to controls in the beta and gamma frequency ranges.
Skosnik et al. 2014	Gamma oscillations (40–59 Hz) during coherent motion perception	Human Subjects (Chronic cannabis users *n* = 34 and cannabis naïve *n* = 23)	---	Gamma power was reduced during coherent motion perception among cannabis users compared to controls. No differences were found between N100 or P200. Cannabis may interfere with the generation of gamma oscillations that may mediate perceptual alterations.

**Table 9 jpm-11-00017-t009:** Clinical Studies of Cocaine on Gamma Oscillations.

Study	Gamma Frequency Range and Type	Subject Information	Method of Administration	Main Findings
Horrell et al. 2010	Evoked and induced gamma power (30–40 Hz)	Human Subjects (Current cocaine abusers *n* = 10)	---	Decreased regional evoked and induced gamma power to non-target and target cues. Induced gamma power to non-target and target cues was reduced globally.

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
