# Peer review of "The Role of Gamma Oscillations in the Pathophysiology of Substance Use Disorders"

_jpm, 2020, doi:10.3390/jpm11010017_

Round 1

Reviewer 1 Report

The present manuscript aims to investigate the impact of alcohol, tobacco, cannabis, cocaine, and amphetamine, on gamma activity, among pre-clinical and clinical populations during acute and chronic exposure and withdrawal states

Authors did a great job gathering relevant papers, however I have some concerns about the methodology.

The first and most important is the absence of risk of bias procedures for both animal and human studies. This procedure is a central component in systematic reviews and under no circumstances can be excluded. Authors must select appropriate risks of bias and quality procedure and include in it tin the manuscript. This outcome might impact the results of the gathered papers as well as conclusions. Risk of bias tables should be included as supplementary material

Second, when did authors made the last update for search?

Third, although PROSPERO register is not mandatory is highly recommended for systematic reviews. This ensures that the methodology is appropriate. Authors should disclose to reviewers their rationale for not including it at any of the systematic review registered protocol (PROSPERO, Cochrane...).

Forth, a PRISMA use statement that it has been followed in the method section is recommended.

Lastly, a pilot search must be included as part of supplementary material

Regarding the results, human and animal are rather mixed jumping from animals to human constantly, mixing adolescents and rats for example. This would be very confusing for readers. I suggest to make different sections for human and  for animals, as well as tables.

A minor language spelling in the inconsistent use of naive and naïve through the manuscript.

Reviewer 2 Report

The authors have reviewed in this paper the role of gamma oscillations
in the pathophysiology of substance use disorders (SUDs), including tobacco,
alcohol, cocaine, cannabis, and amphetamine. Their results indicated that
gamma oscillations appear to be differentially modulated in clinical vs.
preclinical populations, establishing the most complex relationship with
alcohol. It is an excellent review on a relevant topic with clinical
implications, well done from a methodological point of view, and very
exhaustive in describing the results obtained (with additional
summary/tables).

Reviewer 3 Report

This is a very well-written manuscript, in which the authors have assessed a comprehensive review of the literature concerning studies investigating changes in gamma oscillations after substance abuse.

In the results section, authors should avoid including studies assessing electroencephalographic registrations in animals (preclinical studies), since the effects of substance abuse in animals are usually different, compared to human subjects. Also, after eliminating the “preclinical studies”, the manuscript would be “lighter” and easier to read.

In addition, the authors should summarize the “Clinical studies” sections, in order to clarify the reading: instead of disclosing each one of the studies, authors should summarize the results more briefly (e.g., Studies A, B and C showed X results, while studies D, E and F showed mainly Y results).

Reviewer 4 Report

The manuscript by Ramlakhan et al is a concise, well-organized, clearly-written review of the effects of five common drugs of abuse on neural gamma oscillations in human and animal studies. It will be useful to basic researchers and clinicians trying to relate gamma-related neural mechanisms to cognitive symptoms in patients with substance-abuse disorders. 

However, I do have a number of corrections and suggestions for improvement. Below I list them with the relevant line numbers in the manuscript. I have marked the most important ones with an asterisk.

  1. “impaired” would be more unambiguous rather than “modulated” here.
  2. “30 articles…” Figure 1 refers to “title and abstract screening” but it is not clear what the exclusion criteria were. They should be made explicit so the reader can judge whether the winnowing from 322 to 30 articles might have introduced a bias into the reviewed results.
  3. “administered one of nicotine”—one what? Missing word?
  4. “greater sensory gating”—a brief explanation of the sensory gating paradigm would be helpful here since it is also referred to again below.
  5. “but were” should be “but who were”
  6. ASSR should probably be spelled out for readers who may be unfamiliar with the auditory steady state response paradigm.

329 and 333. Again, it would be helpful to briefly describe what it means to have “reduced sensory gating.”

*339-40. “Coherent motion conditions also yielded a significant increase in gamma band activity in chronic users compared to controls.” This sentence appears to contradict the previous sentence, which reports a decrease in gamma in the same condition. 

  1. Here decreases in gamma are interpreted as increased cortical excitability, but at lines 214-15 increased gamma is interpreted as “hyperexcitability.” This should be clarified or made consistent.

*353. “decrease”—this seems to be an error because the following lines describe an increase in gamma synchrony.

  1. “of self-reporting method” should be “of the self-reporting method.”
  2. “as” should be “and.”
  3. Missing word: “locked gamma” should be “locked to gamma.”

409-10. “…acute versus chronic exposure that may be related to cognitive dysfunction…”—not clear what it is that “may be related to” etc.

  1. “…in both directions” is unclear—sketchier than it needs to be.

*419. I thought cannabis led to a decrease in gamma (line 343), so I think the claim here that it increases gamma is wrong.

421-22. “Thus, it appears that alcohol may act differently than other substances.” This statement is made a couple times but doesn’t really seem supported by the results reviewed. For example, cannabis appears to have quite different effects from nicotine…so I don’t necessarily see the justification for treating alcohol as the odd one and the others as similar to each other.

**451-53. “Brain stimulation treatments, such as transcranial direct current stimulation and repetitive transcranial magnetic stimulation, that modulate gamma activity may offer a promising potential approach for targeting working memory.” An important omission from the review is a series of studies demonstrating the effectiveness of non-invasive gamma entrainment therapies on brain pathology and cognitive symptoms in Alzheimer’s model mice, such as:

“Short-Term Effects of Rhythmic Sensory Stimulation in Alzheimer's Disease: An Exploratory Pilot Study” by Amy Clements-Cortes 1 2 3 4, Heidi Ahonen 2, Michael Evans 3, Morris Freedman 4, Lee Bartel 1

Martorell AJ, et al. Multi-sensory gamma stimulation ameliorates alzheimer’s-associated pathology and improves cognition. Cell. 2019 April;177(2):256-271.e22. doi: 10.1016/j.cell.2019.02.014. Epub March 14, 2019.

Adaikkan C, et al. Gamma entrainment binds higher-order brain regions and offers neuroprotection. Neuron. 2019 June;102(5):929-943.e8. doi: 10.1016/j.neuron.2019.04.011. Epub May 7, 2019.

Although they do not relate directly to substance abuse or schizophrenia, these are relevant to the potential for non-invasively modulating gamma activity, and should be cited in the Discussion.

These are all my suggestions; I am not submitting any secret comments to the editor.

Round 2

Reviewer 1 Report

I am satisfied with author's response. Good job